# Deglacial variability of South China hydroclimate heavily contributed by autumn rainfall

Chengfei He [1,2,3✉], Zhengyu Liu [2], Bette L. Otto-Bliesner [4], Esther C. Brady[4], Chenyu Zhu[5,3], Robert Tomas[4], Sifan Gu[6,3], Jing Han[7] & Yishuai Jin[5]

The deglacial hydroclimate in South China remains a long-standing topic of debate due to the lack of reliable moisture proxies and inconsistent model simulations. A recent hydroclimate proxy suggests that South China became wet in cold stadials during the last deglaciation, with the intensification proposed to be contributed mostly by the East Asian summer monsoon (EASM). Here, based on a deglacial simulation in a state-of-the-art climate model that well reproduces the evolution of EASM, winter monsoon (EAWM) and the associated water isotopes in East Asia, we propose that the intensified hydroclimate in South China is also contributed heavily by the rainfall in autumn, during the transition between EASM and EAWM. The excessive rainfall in autumn results from the convergence between anomalous northerly wind due to amplified land-sea thermal contrast and anomalous southerly wind associated with the anticyclone over Western North Pacific, both of which are, in turn, forced by the slowdown of the Atlantic thermohaline circulation. Regardless the rainfall change, however, the modeled $\delta^{18}O_p$ remains largely unchanged in autumn. Our results provide new insights to East Asia monsoon associated with climate change in the North Atlantic.

[1] College of Atmospheric Sciences, Nanjing University of Information Science and Technology, Nanjing, China. [2] Department of Geography, The Ohio State University, Columbus, OH, USA. [3] Open Studio for Ocean-Climate-Isotope Modeling, Pilot National Laboratory for Marine Science and Technology, Qingdao, China. [4] Climate and Global Dynamics Laboratory, National Center for Atmospheric Research, Boulder, CO, USA. [5] Key Laboratory of Physical Oceanography, Ocean University of China, Qingdao, China. [6] School of Oceanography, Shanghai Jiao Tong University, Shanghai, China. [7] Department of Atmospheric and Oceanic Sciences, Peking University, Beijing, China. ✉email: he.1519@osu.edu

E ast Asian monsoon (EAM) is of great importance to the civilization in East China and its surrounding regions. The two major components of the EAM, the warm-wet summer monsoon (EASM) and the cold–dry winter monsoon (EAWM), have been studied extensively in both observations and models during the last deglaciation[1–6]. In particular, the variability of EASM and EAWM in North China is reasonably understood, where the monsoon circulation and rainfall are well simulated in climate models consistent with abundant moisture proxies[2,3,5–7]. However, the deglacial variability of precipitation in South China is much less understood because of the lack of reliable proxies for moisture and substantial inconsistency among climate models[2,8–10]. A recent moisture proxy from Haozhu cave reveals wet cold stadials during the last deglaciation in South China, opposite to the dryness in North China[9]. The wet signal has been proposed to result from the intensification of EASM in South China, as confirmed by model simulations[2,6,9]. In response to the declining Atlantic meridional overturning circulation (AMOC)[11], East China features a dipole hydroclimate response, with a wet South China and a dry North China, associated with reduced low-level southerly monsoon flow, southward migration of high-level westerly jet[2,9,12], and the associated silk-road teleconnection[6]. In contrast, the EAWM in South China, characterized by intensified northerly wind during cold stadials, is out of phase with the EASM due to the land–sea thermal contrast caused by the AMOC[1,4,13].

Relative to EASM and EAWM, the evolution of EAM in other seasons has received much less attention, probably due to the smaller rainfall amount than the summer monsoon[14] in a year (Supplementary Fig. 1). Here we revisit the EAM in the isotope-enabled transient climate experiment (iTRACE) from the last glacial maximum (LGM, -20ka) to the early Holocene (11ka) in the water isotope enabled Community Earth System Model 1.3 (iCESM1.3, Methods). We will focus on the East Asian monsoon rainfall from March to November that covers the majority of the EAM rainfall, with the emphasis on the autumn-monsoon rainfall. Our simulation quantitatively reproduces the evolution of water isotopes and rainfall during the last deglaciation, consistent with cave records across the pan-Asia monsoon region[6], as well as a gradual transition of spring rainfall that follows the orbital precession forcing. In particular, our simulation shows a rainfall response in South China largely consistent with the moisture-proxy record in Haozhu cave[9]. However, this rainfall variability is contributed by the autumn (September–October–November, SON) season as well as the EASM almost equally, highlighting the key role of autumn monsoon in South China hydroclimate during the deglaciation.

## Results and discussion
**Simulated East Asian monsoon.** Overall, the iCESM1.3 is able to simulate the seasonal cycle of the East Asian monsoon rainfall comparable with the modern observation[15], albeit with a weak wet bias (Supplementary Fig. 1a, b; also see He et al.[6] for model validation). The iCESM1.3 also well reproduces the EASM and EAWM evolution as well as the speleothem $\delta^{18}O_c$ (for $\delta^{18}O$ in the cave) during the last deglaciation (Fig. 1a–d). Starting from the LGM, model $\delta^{18}O_c$ undergoes enrichment–depletion–enrichment during the Heinrich Stadial 1 (HS1, ~18–14.5ka), Bølling–Allerød Interstadial (BA, 14.5–12.9ka), and Younger Dryas Stadial (YD, 12.9–11.7ka), consistent with $\delta^{18}O_c$ records in speleothems of Hulu[10] and other caves across the pan-Asia and at Greenland[6,16] (Fig. 1b). Opposite to the amount effect, the summer precipitation in South China (20–35°N,108–120°E) increases in HS1 and YD, but decreases in BA, which largely agrees with the moisture index from Haozhu cave, a proxy for South China monsoon rainfall[9] (Fig. 1c). The modeled EAWM index, defined as the

normalized near-surface (925–1000 hPa) meridional wind strength[1], also intensifies notably in HS1 and YD and weakens in BA, resembling the sea-surface temperature gradient in the South China Sea (SCS) and Ti content at lake Huguang Maar in South China[4,13] (Fig. 1d), which have been considered as proxies for EAWM.

In addition to the EASM and EAWM consistent with other simulations[1] and observations[3,4,9], our model further simulates dramatic rainfall transitions across the period of HS1-BA-YD in South China during the autumn season, which evolves similarly to the summer-monsoon rainfall, and, in particularly, with a response magnitude comparable with the summer rainfall (Fig. 1c and Supplementary Fig. 1c–e). Interestingly, this suggests that, even though the autumn rainfall has a minor contribution to the total rainfall amount in a year, the deglacial change of autumn rainfall can still contribute substantially to the deglacial change of total annual rainfall over South China.

Using sensitivity experiments, the rainfall-evolution responses can be decomposed approximately into those to different climate forcings. This decomposition reveals that the dramatic transitions in autumn rainfall are primarily driven by the meltwater forcing via the variability of AMOC (Fig. 1e and Supplementary Fig. 2b). Greenhouse gas and orbital forcing have little impacts on the autumn rainfall, although both contribute substantially to the summer rainfall (Supplementary Fig. 2a,b). The ice sheet and ocean basin lead to abrupt rainfall changes in autumn at 14ka and 12ka due to the abrupt change of model ocean bathymetry (see Supplementary Text).

In comparison to the striking rainfall responses in fall and summer, the rainfall response in spring (March–April–May, MAM, dominated by that in May) is moderate over South China, and is characterized by a gradual increase toward the late HS1 followed by a decrease toward the early Holocene (Supplementary Fig. 2c). This rainfall, broadly known as spring-persistent rain[17], is driven by the development of southwesterly wind in the lower troposphere in modern-day climate[18]. The deglacial evolution of the spring rainfall correlates well with the southerly monsoon wind, and appears to be regulated predominantly by the local MAM insolation (Supplementary Fig. 2c).

In contrast to South China, however, the precipitation change in North China is determined almost entirely by that in summer, with little response in the autumn and spring season (Supplementary Fig. 2d–f).

**Responses of East Asian summer and autumn monsoons to AMOC.** The mean summer circulation at LGM is characterized by a vigorous Subtropical High over the North Pacific that transports abundant moisture to East Asia by prevailing southerly monsoon flow, leading to the monsoon rainfall (Supplementary Fig. 3a). The global rainfall response in cold stadials, shown as the difference between HS1 and LGM, indicates a southward migration of the Intertropical Convergence Zone as a result of the declining AMOC[19,20] (Supplementary Fig. 3b). Regionally, the EASM appears to show a uniform wet response across South and North China associated with intensified Subtropical High (Supplementary Fig. 3b). This wet signal, however, has distinct seasonal evolution features associated with the migration of the westerly jet and low-level monsoon flow, which are caused by the different responses to meltwater and solar insolation in different seasons (Supplementary Fig. 1e). In late May and June, the intensified South China rainfall is caused mainly by the meltwater cooling, which leads to a southward displacement of the westerly jet and a reduced southerly monsoon flow, a robust response seen in previous studies[6,9,12]. In July, summer insolation forces a strong warming that overwhelms the meltwater cooling; this

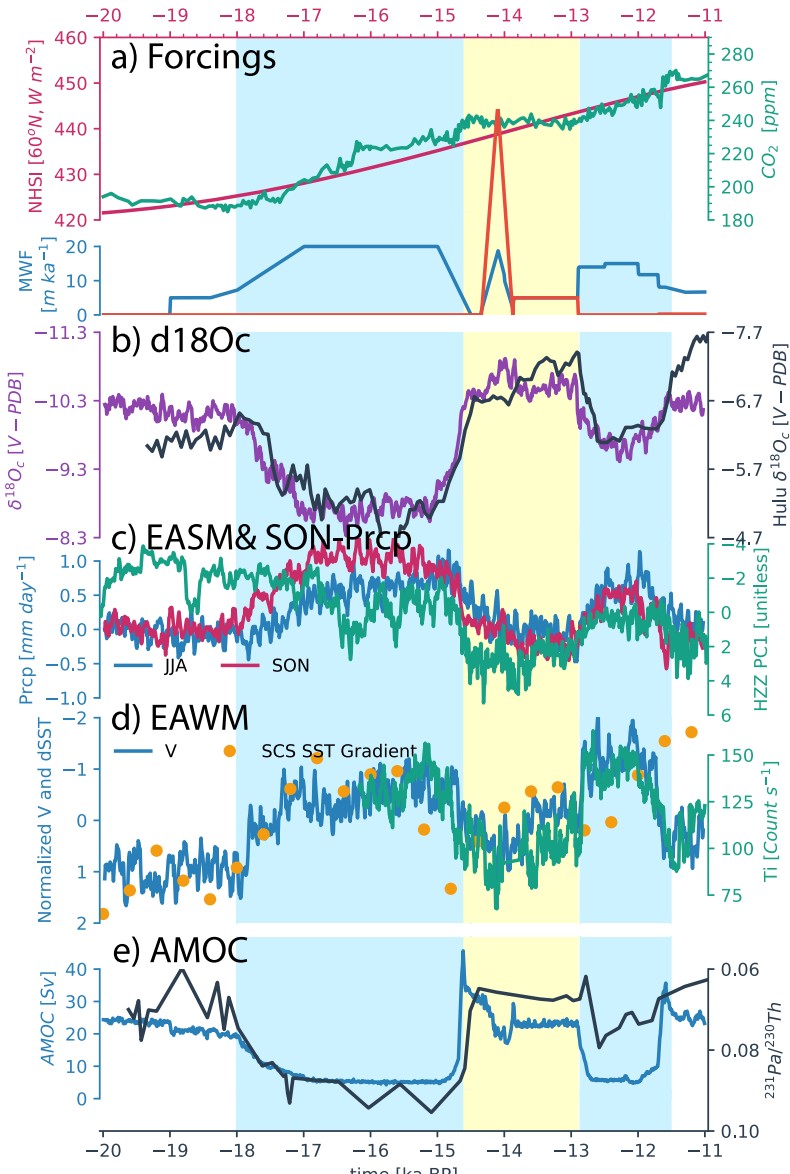

**Fig. 1 Deglacial evolution of East Asian monsoon in iTRACE and paleoclimate proxy. a** Forcing: June insolation at 60°N (red), atmospheric $CO_2$ concentration (green), and meltwater fluxes in the Northern Hemisphere (blue) and Southern Hemisphere (orange); **b** cave $\delta^{18}O_c$ in iTRACE (purple) and observation (black) at Hulu cave[10]; **c** South China [20–35°N, 108–120°E] summer (blue) and autumn (red) precipitation and the leading PC of Mg/Ca, Sr/Ca, and Ba/Ca records observed in Haozhu[9], a proxy for South China monsoon precipitation (green); **d** EAWM index (blue) in iTRACE, SCS SST-gradient index (dots)[13], and sediment Ti content (green) from Lake Huguang Maar[4], proxies for EAWM; **e** simulated AMOC intensity (blue) and observed $^{231}Pa/^{230}Th$ in sediment core GGC5 as a proxy for AMOC intensity (black)[11]. In **c**, the ice sheet and ocean bathymetry effect is removed in autumn precipitation (see supplementary). In **d** the EAWM index is defined as region average [20–35°N, 108–130°E] of the near-surface (925-1000hPa) meridional wind. The SST gradient and EAWM index are normalized for presentation.

warming leads to a northward shift of the westerly jet and intensified southerly monsoonal flow, which transports more moisture to North China and leads to a wet anomaly there[1] (Supplementary Fig. 1e; see Supplementary Text).

In spite of the seemingly similar deglacial rainfall evolution of autumn and summer in South China (Fig. 1c), the patterns of the circulation and their responses differ considerably between the two seasons (Supplementary Fig. 3). In autumn, the Aleutian Low is almost developed in the North Pacific; the East Asian Trough appears associated with a low-level anticyclonic circulation in a growing Siberian High across Eurasia (Supplementary Fig. 3c). Local to East Asia, the low-level northerly winter wind in the East Asian Trough intensifies and starts to take over southerly summer wind in East China, along with the gradual southward retreat of rainfall belt and westerly jet (Supplementary Fig. 1a,b–e and Supplementary Fig. 3c). The prevailing northeasterly wind along the East Asia coast is now fully established and further penetrates into Western North Pacific (WNP), resembling the circulation in EAWM (Supplementary Fig. 3c). In HS1, both the Siberian High and Aleutian Low are intensified (Supplementary Fig. 3d) by the prevalent cooling across the North Hemisphere caused by the meltwater forcing[1]. East China now becomes wet in South China and dry in North China (Supplementary Fig. 3d), a dipole response also seen in observations[9]. This rainfall response is caused by the low-level moisture convergence between the anomalous northerly wind from the mid-latitudes and the

anomalous southerly wind from the south, the former being related to the deepening of the Siberian High and Aleutian Low, while the latter being associated with the anticyclone in the WNP (WNPAC).

The covariability of the precipitation in South China, northerly wind in North China, and WNPAC is also present during the entire deglaciation, as shown in the maximal covariance analysis[21] (MCA, Methods) between SON precipitation and the circulation at 850 hPa. The leading mode, accounting for 62% of total variance, is highly consistent with the response in HS1 (Fig. 2a and Supplementary Fig. 3d) and YD. It is characterized by a meridional dipole response of rainfall, with intensified rainfall in South China and weakened rainfall over WNP extending from

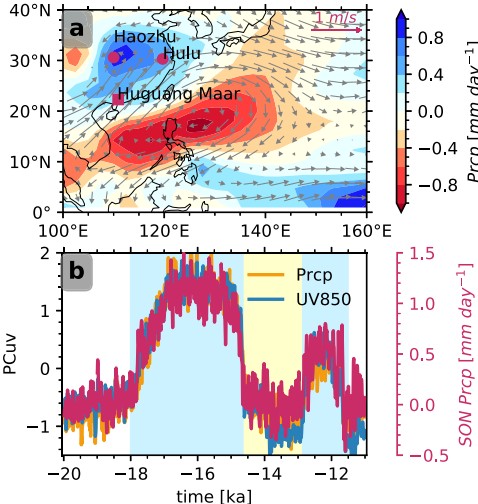

**Fig. 2 Coherence in autumn rainfall and WNPAC. a** Leading mode of the MCA analysis between precipitation (shading) and 850-hPa circulation (vector); **b** time-expansion coefficients of the precipitation (yellow) and circulation (blue), and autumn-precipitation anomaly time series relative to LGM (red). In **b**, the time-expansion coefficients are normalized to unit variance.

SCS to the Philippines Sea (Fig. 2a). Coupled with the reduced rainfall over WNP, the southwest-to-northeast tilted WNPAC persists overhead. The west flank of the anticyclone transports moisture toward South China, where the warm–wet southerly wind converges with the cold–dry northerly wind, producing large-scale precipitation (Fig. 2a). The time-expansion coefficients of rainfall, low-level monsoon flow, and the AMOC are highly correlated with the precipitation responses in autumn, highlighting the coherence in the autumn East Asian rainfall, WNPAC, and the AMOC during the entire deglaciation (Fig. 1e and Fig. 2b). A natural question is, therefore, why WNPAC persists in autumn during the AMOC off-state?

**Enhanced meridional SST gradient maintains the WNPAC in autumn.** From the energic perspective, the anticyclone and descending-motion anomaly over the WNPAC, for example, from LGM to HS1, is caused by an import of negative (i.e., export) moist static energy (MSE) by the low-level northerly wind that overwhelms the net energy-flux input into the atmospheric column. In MSE budget analysis, ascending motion tends to export MSE out of the atmosphere column at the tropopause[22–24], such that any imbalance in MSE may trigger anomalous vertical motion and circulation anomalies. From LGM to HS1, we perform MSE budget analysis (see Methods) over the vicinity of Philippines and SCS north of the warm pool at HS1 and LGM, where the main body of WNPAC resides (box in Fig. 3). The analysis is based on 6-hourly data outputted from two snapshot simulations of iCAM, the atmospheric component of the iCESM1.3, at HS1 and LGM, respectively (see Methods). The iCAM simulations reproduce well the autumn precipitation and circulation in iTRACE (Supplementary Fig. 4a).

The climatology of SON MSE budget terms and vertical motion at LGM are illustrated in Supplementary Figs. 5 and 6. Overall, the whole region features a top–heavy deep convection throughout the atmosphere that vertically advects MSE out of the atmosphere nearby the warm pool (south of 20°N in Supplementary Fig. 5a, Supplementary Fig. 6a1). In HS1, the vertical MSE-advection response, predominantly dominated by dramatic subsidence anomaly (Supplementary Fig. 6a1 and Supplementary

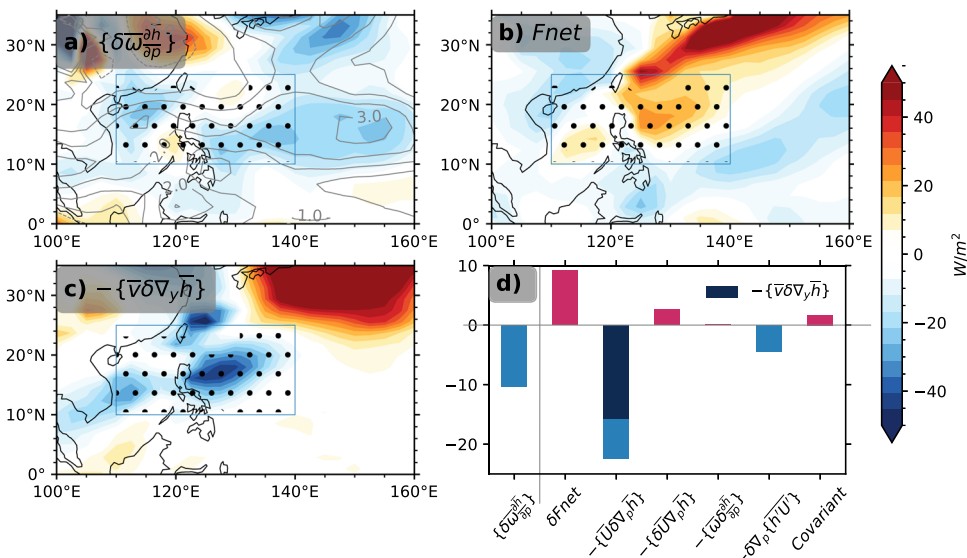

**Fig. 3 MSE budget analysis for the autumn WNPAC. a** Column-integrated vertical MSE-advection anomaly (shading) associated with vertical motion change and vertical $p$-velocity anomaly (contour, Pa/s); **b** net energy source change from top of the atmosphere and surface; **c** column-integrated horizontal MSE-advection anomaly associated with meridional MSE-gradient change; **d** region-averaged anomalous MSE budget terms. In **a–c**, the dots correspond to anomalous descending flow. In **d**, the region is defined as [10–25°N, 110–140°E] as the box in **a–c**. The $-\{\overline{v} \cdot \delta(\nabla_y \overline{h})\}$ (legend, dark blue) dominates the horizontal MSE-advection anomaly $-\{\overline{u} \cdot \delta(\nabla_p \overline{h})\}$, and $-\{\overline{\omega} \cdot \delta\left(\frac{\partial \overline{h}}{\partial p}\right)\}$ is almost zero. See context and Methods for details.

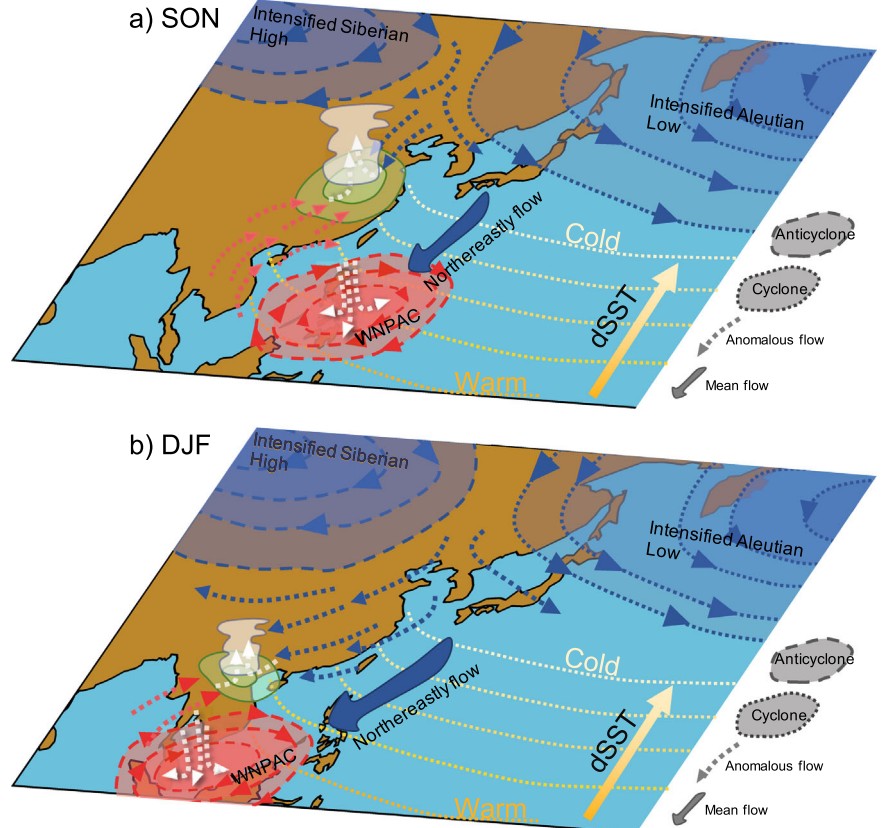

**Fig. 4 Schematic plot for the mechanism of the autumn monsoon rainfall and WNPAC with its evolution in winter. a** Autumn. **b** Winter. In **a**, the rainfall response is mainly forced by the slowdown of the AMOC. The AMOC-induced cooling in the atmosphere deepens the Siberian High and Aleutian Low that strengthens the northerly wind in North China in autumn. Meanwhile, the decline of AMOC also yields intensified meridional SST gradient that cools and desiccates the lower atmosphere along the coast of East Asia. The mean northeasterly wind transports the air of anomalous low MSE to the WNP that produces and sustains the WNPAC. The anomalous northerly wind associated with Siberian High and Aleutian Low converges with anomalous southerly wind associated with the WNAPC in South China, leading to notable precipitation. In **b**, the low-level mean northeasterly wind enhances in winter that shifts the WNPAC southward; the anomalous Siberian High and Aleutian Low also intensifies that makes the anomalous northerly wind prevail in South China.

Fig. 7a), tends to "import" energy into the atmosphere column (Fig. 3d) and is in the first order determined by the net energy-flux change $\delta Fnet$ and the MSE horizontal advection caused by the change of MSE gradient $\left(-\left\{\bar{\boldsymbol{u}} \cdot \delta(\nabla_p \bar{h})\right\}\right.$, Fig. 3d). The increased surface-latent and sensible heat flux imports MSE into the WNP atmosphere, increasing $Fnet (\delta Fnet > 0)$ (Fig. 3d) and favoring convection and anomalous ascending motion (Supplementary Fig. 8c and d). However, this energy import is reduced substantially by the increased outgoing longwave radiation associated with reduced convection and shallowing cloud[25,26] (Supplementary Fig. 8b). More importantly, an even greater amount of MSE is exported by horizontal advection, mostly by the advection on the anomalous meridional MSE gradient in the lower atmosphere $\left(-\left\{\bar{\boldsymbol{u}} \cdot \delta(\nabla_p \bar{h})\right\} < 0\right.$, Fig. 3c, d and Supplementary Fig. 7b). The horizontal export is so strong that it overwhelms the net energy-flux import and leads to a net MSE loss in the atmosphere over WNP. This MSE loss leads to a MSE import by the vertical advection into the atmosphere column $\delta\left\{\bar{\omega}\frac{\partial \bar{h}}{\partial p}\right\} < 0$ by anomalous descending motion (contours in Fig. 3a and dots in Fig. 3c, Supplementary Fig. 7a). The anomalous descending motion further induces anticyclonic anomaly, sustaining the subtropical high in the WNP (Fig. 2a). The anomalous descending zone and the anomalous anticyclone, which spans zonally from east of Philippines to the SCS, fits well with the

anomalous horizontal MSE export tied to enhanced MSE gradient (Fig. 2a, Fig. 3a, c).

Further examination shows that the intensified meridional MSE gradient (Supplementary Fig. 6c2) is caused mainly by the change in moist enthalpy ($C_p T + Lq$), mostly in moisture, in the lower atmosphere associated with the enhanced meridional SST gradient (Supplementary Fig. 9a–c). Given the enhanced meridional MSE gradient (Supplementary Fig. 9a and 6c2), the low-level mean northeasterly wind advects air of anomalous low MSE from mid-latitudes toward the Philippines Sea and SCS (Supplementary Fig. 9a–c and Supplementary Fig. 6c), importing negative (i.e., export) MSE, suppressing convection, and therefore sustaining an anomalous anticyclone (Fig. 4a, Fig. 2a and Supplementary Fig. 4a). In comparison, there is little zonal gradient in MSE, leading to little responses in zonal MSE advection over WNP (Supplementary Fig. 6b2, 6b3), in spite of an intensified high-level westerly in low latitudes (Supplementary Fig. 8a and 6b1).

The anticyclonic response may be further enhanced by a positive feedback between the reduced convection and diabatic cooling, as reduced convection and rainfall release less latent heat to the atmosphere, equivalent to net MSE export, which favors shallowing convection and deepening anticyclone[27]. Later, in winter (December–January–February, DJF), the center of WNPAC moves to the south of the Indo-China Peninsula (Supplementary Fig. 10a), driven by the further intensification of

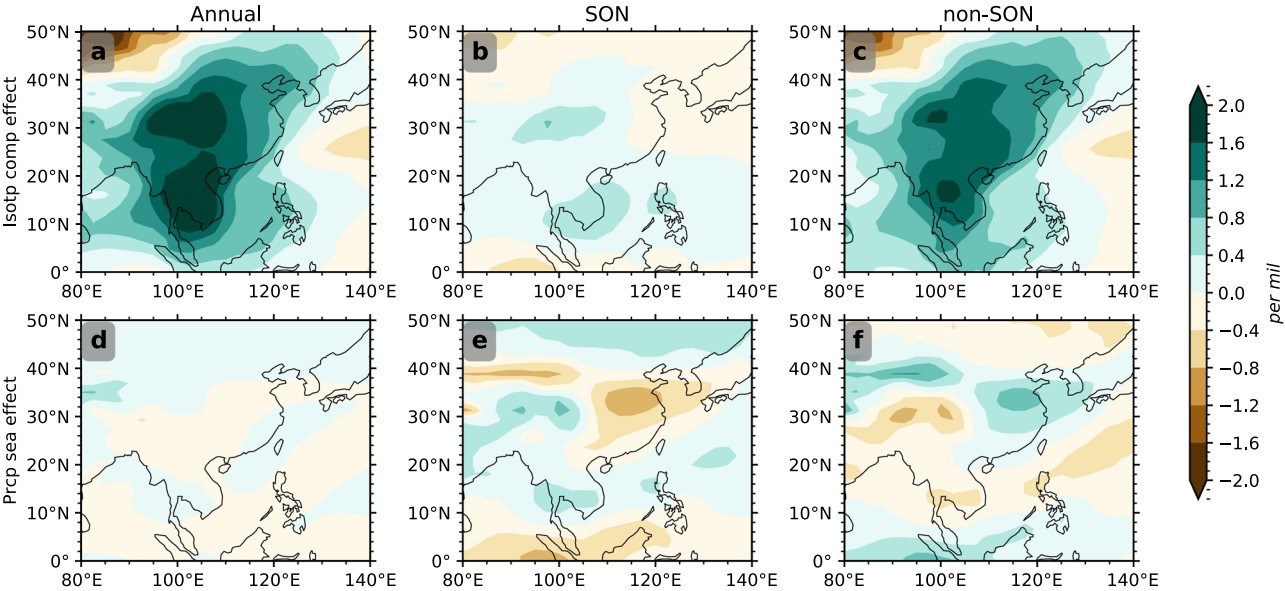

**Fig. 5 $\delta^{18}O_p$ response between HS1 and LGM (HS1–LGM).** a Annual isotopic composition effect. **b, c** As in **a** but for portions of SON and non-SON. **d** Annual precipitation-seasonality effect, **e, f** as in **d**, but for portions of SON and non-SON.

mean northeasterly wind over anomalous low-level MSE along with SST gradient (Supplementary Fig. 10b), conceding South China to the northerly wind as enhanced EAWM (Fig. 4b and Supplementary Fig. 10) due to thickening of Siberian High[1].

**Influence of East Asian autumn monsoon on the speleothem $\delta^{18}O_c$.** In spite of its major contribution to the annual rainfall in South China, the dramatic change of autumn precipitation, in particular in HS1 and YD, has little imprint on the annual precipitation weighted $\delta^{18}O_p$ there. This can be seen from the decomposition of the $\delta^{18}O_p$ change from LGM to HS1 into the components associated with precipitation seasonality and isotopic composition (Fig. 5) (see Method). As anticipated, the increase of autumn rainfall yields a depletion response (Fig. 5e), with the spatial pattern resembling the rainfall anomaly (Fig. 2). Physically, the increase in SON precipitation in cold stadials increases the precipitation weight in SON $(\Delta\left(\frac{P_{SON}}{P}\right)>0)$, such that the precipitation-seasonality effect leads to a depletion $(\Delta\left(\frac{P_{SON}}{P}\right)\delta^{18}O_{SON}<0)$ of the negative $\delta^{18}O_{SON}$. However, this depletion is nearly perfectly compensated by the decreased weight of non-SON precipitation $(\Delta\left(\frac{P_{non-SON}}{P}\right)<0)$, because the total weight change remains zero $\sum_{i=1}^{12}\Delta\left(\frac{P_i}{P}\right)=0$, leaving little net annual effect in South China in HS1 (Fig. 5d–f) and the whole last deglaciation (Fig. S4 in ref. [6]). Furthermore, the autumn rainfall has little effect on the isotopic composition in the SON (Fig. 5b), as the enrichment from LGM to HS1 is caused almost completely by the upstream enrichment and local recycling in the summer[6,8]. Similar to the autumn, the change in the spring-precipitation seasonality is negligible in South China either from LGM to HS1 (Fig. 5d) and during the last deglaciation[6], even though $\delta^{18}O_p$ is to a large extent contributed by spring rainfall in modern-day seasonal cycle in observation[28].

**Twisted relationship between EASM and EAWM.** The iTRACE shows an intensified winter-monsoon wind in cold stadials, consistent with the prior TRACE simulation[1]. This intensified winter-monsoon wind is also consistent with the EAWM proxy of

South China Sea SST gradient[13] and the Ti records of lake Huguang Maar[4] (Fig. 1d), although the proxy representation of the latter remains controversial[29,30]. The relation between EAWM and EASM, however, is subject to the definition of the intensity of the EASM, because of the opposite responses of rainfall between North and South China during the summer. Since an increased monsoon rainfall in South China is accompanied by a reduced southerly monsoon wind[2,6,9], the EAWM is out-of-phase with the EASM, if the EASM intensity is defined in southerly monsoon wind as often used in present climate study[2]. However, the EAWM is in-phase with the EASM if the EASM intensity is defined in local rainfall, opposite to the speculations in Yancheva et al.[4]. This twisted relationship primarily results from the negative correlation of summer southerly wind and precipitation in South China on millennial[9], and possibly also orbital[6] timescales. Physically, a progressive seasonal penetration of southerly wind brings abundant moisture for South and North China rainfall. In millennial events such as HS1 and YD, the penetration of southerly wind is weakened by the increased land–sea thermal contrast[1] and southward shift of westerly jet[12]. As such, the monsoon rainbelt is mostly "trapped" in South China, leading to a dipole hydroclimate response between South China and North China. Therefore, one should always be specific in referring to the intensity of EASM.

**Comparison with the influence of WNPAC on other time-scales.** It has been suggested that the WNPAC influences the EAWM and EASM on a variety of time scales, including the interannual El Nino-Southern Oscillation (ENSO)[25,31–35], multimillennial oscillations in the Holocene, and 50-ka oscillations in the late Quaternary[36]. The persistence of the WNPAC is usually in winter and involves an ENSO-like SST pattern: warming in the tropical eastern Pacific and cooling in the west warming pool. The warm eastern Pacific could trigger two Rossby wave cyclonic anomalies to the west[37], and the northern branch of the cyclones advects dry and cold air to WNP, suppressing local convection and maintaining the WNPAC[25]. Local to WNP, the cold western Pacific inhibits the local convection directly and amplifies the WNPAC via feedbacks in surface wind-evaporation SST[33,38,39].

Here we show that the meridional SST change associated with the slowdown of the AMOC, rather than zonal SST change in the

ENSO, could also produce and maintain the WNPAC by coupling with mean northeasterly monsoon flow in autumn (Fig. 4a) and cause excessive rainfall in South China. In addition, the intensified mean northeasterly monsoon over the SST anomaly shifts the center of WNPAC in winter (Fig. 4b). The anticyclonic circulation response during the deglacation over WNP seems to be robust, as it is also confirmed in TRACE21ka transient simulation[20] (Supplementary Fig. 4b). Interestingly, while the rainfall amount changes significantly during deglaciation in autumn, the $\delta^{18}O_p$ change across the East Asia is determined predominantly by the summer rainfall[6]. This implies a divergence between precipitation and water isotope $\delta^{18}O_p$, not only in space[6], but also in seasonal rainfall contribution. More independent precipitation proxies and seasonally resolved speleothem $\delta^{18}O_c$ records[40,41] need to be developed in the future to further our understanding of the seasonality of EAM in the past.

## Methods

### iTRACE and iCESM1.3.
The iTRACE simulation largely mimicking the previous transient simulation: TRACE21ka, is performed in the state-of-the-art iCESM1.3 but with water isotope enabled[42–45]. The embedding water isotope enables us to compare the modeled and observational water isotope directly to understand the past climate change. The iCESM1.3 is composed by the community atmosphere model (CAM5.3)[46], Parallel Ocean Program, version 2 (POP2)[47], Los Alamos Sea Ice Model, version 4 (CICE4)[48], and Community Land Model, version 4 (CLM4)[49]. The resolution of atmosphere and land is about nominal 2° resolutions (1.9° in latitude and 2.5° in longitude), with 30 vertical levels in the atmosphere; the resolution of ocean and sea ice is nominal 1° horizontal resolutions (gx1v6), with 60 vertical levels in the ocean. Our iCESM1.3 could well reproduce the present-day physical climate, including the seasonal cycle of EAM, similar to CESM1.2[50]. From spring to early summer, the rainfall center mostly stays at South China and migrates northward in the mid-summer to North China, following the burst of low-level southerly wind (Supplementary Fig. 1a,b). In autumn, the low-level circulation switches from southerly wind associated with EASM to northerly wind associated with EAWM, while the rainfall belt, in turn, retreats southward, with the intensity significantly weaker than in summer (Supplementary Fig. 1a, b). In winter, the East Asian is prevailed by the northerly wind (Supplementary Fig. 1a, b).

Starting at 20ka, three parallel experiments (ICE, ICE+ORB, and ICE+ORB +GHG) with factorized forcings (ICE: ice sheet (ICE-6G)[51] and KMT (ocean bathymetry), ORB: orbital forcing, GHG: greenhouse gas)[52–54] were branched from an around 1000-year spinup run at LGM. Our LGM spinup shows that the global mean temperature is about 7 °C cooler than the preindustrial, the AMOC is about 25 Sv close to present-day observation[55], but with a shallower depth. At 19ka, the baseline iTRACE simulation (ICE+ORB+GHG+MWF) was branched from the ICE+ORB+GHG run, with meltwater forcing (MWF) largely similar to that in TRACE21ka (Fig. 1a). The $\delta^{18}O$ values in the meltwater were prescribed according to ref. [56] and the $\delta D$ values in the meltwater were the $\delta^{18}O$ values multiplied by a factor of 8. During the simulation, we changed the ice sheet per thousand years (19ka, 18ka, 17ka,…) and modified the ocean bathymetry twice at 14ka and 12ka based on ICE-6G[51]. Please refer to ref. [57] for more technical details.

By comparing the factorized-forcing simulations, the role of each forcing is approximately obtained. For example, (ICE + ORB) – ICE reveals the role of orbital forcing.

### iCAM snapshot simulations.
Two snapshot experiments were set up in the iCAM5.3, the atmospheric model in iCESM1.3, using the SST, sea ice, continental ice sheet, greenhouse gas, and orbital parameters at LGM (20ka) and HS1(15.5ka) respectively. Each run was integrated for 40 years, and the last 20 years' data were used for analysis. For MSE-budget analysis, physical fields, including wind, temperature, moisture, and geopotential height, are saved per 6 h. It turns out that the iCAM snapshot simulation could well represent the iTRACE climate and water isotope at LGM and HS1[6]. The precipitation and circulation responses in SON between HS1 and LGM confirm the first-mode MCA loadings (Supplementary Fig. 4a and Fig. 2).

### MCA analysis.
In present study, we use the SON precipitation as the left field and the UV850 as the right field at the domain of [0–60°N, 90–160°E]. It turns out that there exist two major modes. The first and second mode explain 62 and 23% variances, respectively. The PCs of the second mode correspond to the responses to the greenhouse gas, and the magnitude of loadings is small. MCA codes are available at https://github.com/Yefee/xMCA.

### MSE-budget analysis.
We mostly follow Hill et al.[24] to conduct the MSE-budget analysis, while a similar approach is performed in ref. [25] to examine the WNPAC as

a response of ENSO. Neglecting the small contribution from kinetic energy, the equation for monthly mean atmospheric column-integrated MSE can be written in the pressure coordinate as

$$\left\{ \overline{\omega} \frac{\partial \overline{h}}{\partial p} \right\} \approx \overline{F}_{net} - \left\{ \overline{\boldsymbol{u}} \cdot \nabla_p \overline{h} \right\} - \left\{ \nabla_p \cdot (\overline{\boldsymbol{u'}h'}) \right\} - \{\partial_t \overline{h}\}. \tag{1}$$

The MSE is denoted by $h$, $h = C_p T + Lq + gz$, where $C_p$, $g$, and $L$ are specific heat of air, gravitational constant, and latent heat of vaporization of water, and $T$, $z$, and $q$ are temperature of air, geopotential height, and moisture concentration, respectively. $\omega$ denotes as the $p$-velocity and $\boldsymbol{u} = (u, v)$ is zonal and meridional velocity. The overbar (¯) is the monthly mean; the prime (′) is the transient eddy from the monthly mean; $\{\cdot\} \equiv \int_0^{p_s} \cdot (\frac{dp}{g})$ is the column accumulation, with $p_s$ as the surface pressure.

The first term on the right-hand side (rhs) of Eq. (1) represents the net energy source into the atmosphere from surface (sfc) and top of atmosphere (TOA), such that $F_{net} = F_{TOA} - F_{sfc}$. At TOA, $F_{TOA}$ is the net difference between the incoming shortwave radiation and outgoing longwave radiation. At the surface, $F_{sfc}$ is composed by upward longwave radiation from the surface and downward shortwave radiation, and sensible and latent heat flux. The second term on the rhs denotes mean horizontal MSE advection $-\left\{ \overline{\boldsymbol{u}} \cdot \nabla_p \overline{h} \right\}$, dominated by large-scale rotational flow[24]. The third term $-\left\{ \nabla_p \cdot (\overline{\boldsymbol{u'}h'}) \right\}$ is transient eddy convergence, and the last term $\{\partial_t \overline{h}\}$ is the local MSE tendency, which is usually small for both climatology and anomaly in our calculation and other peer studies[24,58,59]. We neglect it for simplicity. Terms on the rhs in a positive sign mean MSE import into the atmosphere and vice versa. For example, a positive $F_{net}$ means the atmosphere gains energy from either the TOA or surface through heat fluxes. If other terms are kept constant, the MSE tendency $\{\partial_t \overline{h}\}$ would increase.

To infer the vertical velocity, the MSE-budget terms are arranged as in Eq. (1), such that the contribution of each term to the vertical MSE advection $\left\{ \overline{\omega} \frac{\partial \overline{h}}{\partial p} \right\}$ can be easily identified. Opposite to the sign definition on the rhs, positive $\left\{ \overline{\omega} \frac{\partial \overline{h}}{\partial p} \right\} > 0$ implies ascent flow ($\overline{\omega} < 0$) and MSE export at the tropopause, and vice versa. In our present study, over vicinity of Philippines and SCS (box in Fig. 3 and Supplementary Fig. 5), the vertical MSE advection $\left\{ \overline{\omega} \frac{\partial \overline{h}}{\partial p} \right\}$ exports MSE out of the atmosphere column at the upper troposphere south of 20°N that is primarily balanced by the import of net energy source $F_{net}$ (Supplementary Fig. 5a, b and Supplementary Fig. 6a3), a classical picture of tropical convection[22]. In 20–25°N, together with the $F_{net}$, the vertical advection imports MSE into the atmosphere column due to shallow convection[60], compensating the horizontal exporting by the low-level meridional advection (Supplementary Fig. 5 and Supplementary Fig. 6c3). Nevertheless, the WNP region overall features a deep convection, as shown by the domain-averaged vertical motion profile at LGM (Supplementary Fig. 6a1).

Neglecting the tendency term, the responses of MSE-budget terms between HS1 and LGM could be decomposed as

$$\begin{aligned} \left\{ \delta\overline{\omega} \frac{\partial \overline{h}}{\partial p} \right\} + \left\{ \overline{\omega}\delta\left(\frac{\partial \overline{h}}{\partial p}\right) \right\} + \left\{ \delta\overline{\omega}\delta\left(\frac{\partial \overline{h}}{\partial p}\right) \right\} &\approx \delta\overline{F}_{net} \\ -\left\{ \delta\overline{\boldsymbol{u}} \cdot \nabla_p \overline{h} \right\} - \left\{ \overline{\boldsymbol{u}} \cdot \delta(\nabla_p \overline{h}) \right\} - \left\{ \delta\overline{\boldsymbol{u}} \cdot \delta(\nabla_p \overline{h}) \right\} &- \delta\left\{ \nabla_p \cdot (\overline{\boldsymbol{u'}h'}) \right\} \end{aligned} \tag{2}$$

where $\delta$ denotes the difference between HS1 and LGM (HS1 - LGM). The responses of vertical advection $\delta\left\{ \overline{\omega} \frac{\partial \overline{h}}{\partial p} \right\}$ consist of advection change due to vertical motion $\left\{ \delta\overline{\omega} \frac{\partial \overline{h}}{\partial p} \right\}$ (dynamic part), gross stability $\left\{ \overline{\omega}\delta\left(\frac{\partial \overline{h}}{\partial p}\right) \right\}$ (thermodynamic part), and the combination of both $\left\{ \delta\overline{\omega}\delta\left(\frac{\partial \overline{h}}{\partial p}\right) \right\}$ (covariant). As shown in Supplementary Fig. 7a, the vertical motion change term dominates the total vertical MSE advection $\delta\left\{ \overline{\omega} \frac{\partial \overline{h}}{\partial p} \right\}$ throughout the atmosphere. Similarly, the horizontal MSE-advection anomaly $-\delta\left\{ \overline{\boldsymbol{u}} \cdot \nabla_p \overline{h} \right\}$ is composed of the dynamic part $-\left\{ \delta\overline{\boldsymbol{u}} \cdot \nabla_p \overline{h} \right\}$ associated with horizontal wind change, the thermodynamic part $-\left\{ \overline{\boldsymbol{u}} \cdot \delta(\nabla_p \overline{h}) \right\}$ associated with MSE-gradient change and the covariant part $-\left\{ \delta\overline{\boldsymbol{u}} \cdot \delta(\nabla_p \overline{h}) \right\}$. The horizontal MSE-advection anomaly $-\delta\left\{ \overline{\boldsymbol{u}} \cdot \nabla_p \overline{h} \right\}$ is dominated by the low-level thermodynamic $-\left\{ \overline{\boldsymbol{u}} \cdot \delta(\nabla_p \overline{h}) \right\}$ change in the present study (Supplementary Fig. 7b).

Technically, we exactly followed the scheme in Hill et al.[24]. We first corrected the mass and MSE imbalance by assuming the spurious velocity barotropic and irrotational. Afterward, each term was calculated at the native hybrid sigma-pressure grid. Details about the correction and calculation could be found in Appendix from ref. [24].

### Decomposition of precipitation-weighted $\delta^{18}O_p$.
The interpretation and implication of speleothem $\delta^{18}O_c$ on hydroclimate across the East Asian monsoon regions have been long debated[2,8,10,61], which mostly stems from the complicated processes

influencing the record of speleothem $\delta^{18}O_c$. In general, a change in speleothem $\delta^{18}O_c$ is influenced by a change in precipitation weighted $\delta^{18}O_p$ (for $\delta^{18}O$ in precipitation) and the processes in the karst system that are empirically constrained by cave temperature[62]. The change in precipitation weighted $\delta^{18}O_p$, consisting of precipitation-seasonality effect and isotopic composition effect (see Eq. 3 below), has been suggested the dominating component in forging the speleothem records[6,8].

In particular, an increase in precipitation seasonality in subtropics would deplete the speleothem $\delta^{18}O_c$, as an increase in summer precipitation (or a decrease in winter precipitation, or the combination of both) acts to accumulate more light-summer (or less heavy-winter) isotope in speleothem due to amount effect. This effect is evident in South China from a list of records ranging from the last several decades to the Holocene[28,40,41,63]. The isotopic-composition effect is usually more complicated due to the complexity in the processes of water cycle. A change of isotopic composition may originate from redistribution of source water vapor associated with a reorganization of monsoon circulation[64–66], source water vapor $\delta^{18}O$ caused by a change in SST, en route depletion due to a change in Rayleigh distillation and rainout[6,8], and local condensation because of a change in the intensity of convection.

The $\delta^{18}O_p$ responses in East Asia could be decomposed as follows

$$\Delta\delta^{18}O_p = \sum_{i=1}^{12}\left(\frac{\bar{P}_i}{\bar{P}}\right)\Delta\delta^{18}O_i + \sum_{i=1}^{12}\delta^{18}\bar{O}_i\Delta\left(\frac{P_i}{\bar{P}}\right) \qquad (3)$$

where $i$ is calendar month, ranging from 1 to 12, $\bar{P}$ is annual precipitation accumulation, the sum of $P_i$. The total $\Delta\delta^{18}O_p$ is composed by the change in the value of the isotope $\Delta\delta^{18}O_i$ in precipitation (isotopic composition effect) and the change associated with the precipitation weight $\Delta\left(\frac{P_i}{\bar{P}}\right)$ (precipitation seasonality effect), with $i$ as calendar month. In subtropics and tropics such as Asian monsoon regions, an increase in precipitation seasonality corresponds to a depletion effect as the $\delta^{18}O_{JJA}$ in summer rainfall is usually low. However, in high latitudes such as Greenland and the Antarctic, an increase in precipitation-seasonality would enrich the ice core $\delta^{18}O$ due to the temperature effect[16,67].

The two subcomponents $\sum_{i=1}^{12}\left(\frac{\bar{P}_i}{\bar{P}}\right)\Delta\delta^{18}O_i$ and $\sum_{i=1}^{12}\delta^{18}\bar{O}_i\Delta\left(\frac{P_i}{\bar{P}}\right)$ could be further portioned into the SON ($i = 9, 10, 11$) and non-SON ($i \neq 9, 10, 11$) parts, as shown in Fig. 5. Similarly, the precipitation seasonality effect could also be portioned into MAM and non-MAM parts, with the annual total unchanged (Fig. 5d).

## Data availability

All model data supporting our findings are archived at https://zenodo.org/record/4462948. Paleoclimate proxy data could be found from NOAA (https://www.ncdc.noaa.gov/data-access/paleoclimatology-data).

## Code availability

The code for the present study is available in XCESM (https://github.com/Yefee/xcesm) and xMCA repository[68] (https://github.com/Yefee/xMCA, MCA analysis).

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

## Acknowledgements

We thank Dr. Jesse Nusbaumer for helpful discussion on the latent heat flux in CESM. This work is supported by Chinese NSFC 41630527, U.S. NSF 1810682, 1810681, and Qingdao Pilot National Laboratory for Marine Science and Technology. The CESM project is supported primarily by the National Science Foundation (NSF). This material is based on work supported by the National Center for Atmospheric Research, which is a major facility sponsored by the NSF under Cooperative Agreement No. 1852977. Computing and data-storage resources, including the Cheyenne supercomputer (doi: 10.5065/D6RX99HX), were provided by the Computational and Information Systems Laboratory (CISL) at NCAR.

## Author contributions

C.H. and Z.L. conceived this study. C.H. performed the analysis and wrote the paper with Z.L.. Z.L. and B.O.B designed the experiments. C.H. and S.G. developed model code. C.H., E.C.B., R.T. and C.Z. performed the experiments. J.H. and Y.J. contributed to the analysis of spring rainfall. All authors discussed the results and contributed to the paper.

## Competing interests

The authors declare no competing interests.
