## [Peer Review File · Nature Communications]

Deglacial variability of South China hydroclimate heavily contributed by autumn rainfallREVIEWER COMMENTS

Reviewer #1 (Remarks to the Author):

He et al. presents a new set of transient fully coupled climate model simulations through the last deglaciation. The authors use results from these simulations to interrogate how changes in climate forcings influenced the temporal and spatial variability of water isotopes and rainfall amounts across China. What has emerged in the recent literature is that while the $\delta^{18}\text{O}$ of (in particular) speleothem records are more or less homogenous across China through the deglaciation, hydroclimate proxies suggest an antiphased response in rainfall amounts between northern and southern China in response to changes in the Atlantic Meridional Overturning Circulation (AMOC). Yet, what hasn't been so clear, until now, is how the autumn/winter monsoon varied in the past, and to what degree changes during these seasons can contribute to the annual hydroclimate variability witnessed in the proxy records. The authors find coeval changes in both summer and autumn/winter monsoon rainfall changes through the deglaciation in southern China, that is antiphased with the response in northern China. Interestingly, their model results show that this increase in autumn rainfall during Heinrich Stadial 1 and Younger Dryas was, similar to the summer monsoon, likely tied to the AMOC changes and its impact on the meridional SST gradient, which acted to "maintain the Western North Pacific Anticyclone (WPNAC) in autumn". Some caveats of this work are a lack of proxy-model comparisons, but there are a lack of high resolution winter monsoon records so this is not the fault of the authors. However, I do think the authors could have explained in some more detail how their results help to reconcile ongoing debates about what the $\delta^{18}\text{O}$ speleothem records are exactly telling us. And, are the records in northern China, for example, telling us something different from those in southern/central China? We know the $\delta^{18}\text{O}$ in the speleothems across China (and beyond) look very similar through the deglaciation. Is this due to a common mechanism related to what is happening 'upstream' of the summer monsoon moisture source? Or, is it also related to what is happening in autumn/winter, as this study shows in terms of rainfall amount.

The paper is overall well written, and the interpretation of the results appear robust. Hence, I recommend this paper be published once the authors have addressed some minor comments/questions outlined below:

Specific comments:

Line 86-87: The iTRACE model shows increased winter monsoon winds during HS1-YD coeval with the summer monsoon winds. This is interesting and indeed an important finding given that the highly-cited Yancheva et al. (2007) interpreted the opposite in their sediment record. It might benefit the reader to be made aware of this apparent contradiction between the new model results and Ti lake records. I am aware of the controversy as I am involved in this type of paleo work, though I am not sure whether the lay reader will be aware of this. This is important because these new model results shed new light on the synchronicity of the two major monsoon systems, and in my mind, settle once and for all the interpretation of this record.

How do these results fit in with the recent model of Chiang et al. (PNAS, 2020)?

Recently, Chiang et al. (2020; PNAS) demonstrated, using an isotope-equipped climate model constrained by large-scale atmospheric reanalysis fields, that year's corresponding with more enriched $\delta^{18}\text{O}$ of precipitation have a reduced summer seasonality between the months of June and October. Specifically, these authors showed that years with more enriched (depleted) $\delta^{18}\text{O}$ of precipitation have a summer-winter monsoon transition that is less (more) pronounced. The results presented in the iTRACE model are consistent with Chiang et al's results in that both show no "amount effect" in central China, but to what extent do the results here align or not align with those of Chiang et al. (2020)? Obviously, this is not a complete apples-to-apples comparison given that the He et al. manuscript is focused on the deglaciation and Chiang et al. paper is the modern climate, but I am just curious if the authors have considered this mechanism because I did not see much reference to the

westerlies crossing the Tibetan Plateau, and particularly how the timing of this crossing could affect precipitation isotopes in central China during the summer-winter monsoon transition. There were hints of the jet transition hypothesis being mentioned (for example, Lines 107-110), but it just wasn't clear to me how the Chiang model fits in with these new data. Maybe I am missing something.

What are the implications for these results in terms of our understanding of how/why the $\delta^{18}\text{O}$ records across China are mostly synchronous through the last deglaciation despite significant changes in both summer and autumn rainfall amounts? Since the paper of Pausata et al. (2011), the general consensus in the literature has been that the $\delta^{18}\text{O}$ in speleothems reflects more regional atmospheric circulation changes, such as upstream rainout and shifting moisture sources, rather than local rainfall amount. If the model results in this manuscript are accurate, showing that autumn rainfall changes may account for a large proportion of the annual variability through HS1/YD (in addition to summer), then wouldn't this seasonal contribution of autumn rainfall also contribute to the annual $\delta^{18}\text{O}$ precipitation signal recorded in the speleothems? Is it because the patterns of both the summer and autumn rainfall changes through HS1/B-A/YD are similar, at least in central China, that you have such a homogeneous signal across the region through these events? If this is the case, then can we assume that $\delta^{18}\text{O}$ source changes (as proposed by Hu et al. 2020, albeit over orbital and not millennial timescales) may not be the main driver of the $\delta^{18}\text{O}$ changes on orbital and millennial timescales? However, as stated in the paper, there are large seasonal differences in the response of hydroclimate to changes in the AMOC between central and northern China. Thus, can these spatial and temporal changes in rainfall through the deglaciation observed in the model help to explain the speleothem $\delta^{18}\text{O}$ changes through time?

How do the authors reconcile the disparity in rainfall response between the YD and HS1 in northern China? The proxy (particularly lake) records by and large indicate that HS1 was drier in northern China, similar to the YD (see Fig. 3 and references therein in Zhang et al., 2018, Science). In Fig. 5B of He et al., 2021, the first MCA mode for summer rainfall clearly shows a dipole pattern between central and northern China. Given very similar forcings between HS1 and the YD, any explanation for why the model does not reproduce the proxy results in northern China during HS1?

Minor comments:

Line 30: Should be "reliable moisture proxies"

Figure 1a: "Frocings" should read "Forcings"

References

Chiang, John C. H., Michael J. Herman, Kei Yoshimura, and Inez Y. Fung. "Enriched East Asian Oxygen Isotope of Precipitation Indicates Reduced Summer Seasonality in Regional Climate and Westerlies." *Proceedings of the National Academy of Sciences*, June 12, 2020, 201922602. <https://doi.org/10.1073/pnas.1922602117>.

Hu, Jun, Julien Emile-Geay, Clay Tabor, Jesse Nusbaumer, and Judson Partin. "Deciphering Oxygen Isotope Records From Chinese Speleothems With an Isotope-Enabled Climate Model." *Paleoceanography and Paleoclimatology* 34, no. 12 (December 2019): 2098–2112. <https://doi.org/10.1029/2019PA003741>.

Reviewer #2 (Remarks to the Author):

The deglacial hydroclimate in eastern China, especially in southern China, remains in debates because of a lack of reliable paleorainfall proxy and inconsistent model simulations. Based on a deglacial simulation of an isotope-enabled Transient Climate Experiment (iTRACE) performed in the iCESM 1.3, the authors proposed that intensified hydroclimate in South China is also heavily contributed by the autumn rainfall, which are response to the slowdown of the North Atlantic thermohaline circulation

(AMOC). It was suggested that the excessive rainfall in autumn results from the convergence between anomalous northerly wind due to amplified land-sea thermal contrast and anomalous southerly wind associated with an anticyclone at Western North Pacific. As far as I'm concerned, the dynamic interpretation is reasonable based on moist static energy (MSE) budget analysis. The manuscript was well written and I suggest that it can be accepted if my concerns could be addressed/considered.

Here are my questions for the manuscript:

1. In Supplementary figure 1e, we can also find a considerable amount of spring rainfall in March-May. If the authors want to use the precipitation seasonality to explain wet climate but high $\delta^{18}\text{O}$ values in southern China during the off-state of the AMOC, why the spring rainfall was not considered/discussed in this study? Based on instrumental data (1951-2010 CE), some studies suggested that the spring (MAM) rainfall amount is equivalent to the summer monsoon (JJA) rainfall amount in southeastern China, especially in the region of the "spring persistent rain" (Wan and Wu, 2009), which may contribute ~50% to amount-weighted annual $\delta^{18}\text{O}$ values (Zhang et al., 2020). A new Holocene speleothem $\delta^{18}\text{O}$ record from southeastern China also suggested that the spring rainfall may significantly influenced variations in hydroclimate and $\delta^{18}\text{O}$ in southeastern China (Zhang et al., 2021). It will be perfect if the authors could quantify or exclude the effect of spring rainfall on the variations in both hydroclimate and $\delta^{18}\text{O}$ in southeastern China.

2. From the full manuscript, the main experiments and evidences were obtained on the basis of differences between H1 and LGM. I am curious about the difference between YD and BA, which may better show the hydroclimate changes during the last deglaciation. Please point out it if the authors think it's not necessary to show/analysis the differences between the YD and BA or the comparison between H1 and LGM is enough.

3. A tripole pattern of summer monsoon rainfall (dry-wet-dry) over eastern China can be recognized in the observations and model simulations (e.g., Zhang et al., 2018), and the Haozhu cave is located in the middle-lower reaches of the Yangtze River Valley (~35°N). It was suggested that a wetter central eastern China during North Atlantic cooling episodes (the weak or off-state of the AMOC) based on the multiproxy speleothem record of Haozhu cave and model simulation (e.g., Zhang et al., 2018). In this manuscript, the eastern China was divided into North (37-50°N, 108-120°E) and South (20-35°N, 108-120°E) China, in other words, a dipole pattern of rainfall in eastern China. In addition, from the supplementary figure 1e, we do see the autumn rainfall is equivalent to the summer rainfall at the latitude of Haozhu cave (~35°N), however, there is high spring rainfall but low autumn rainfall in the region of 20-28°N. How to reconcile this differences? The authors may reconsider the division of eastern China as North and South China or North, Central-east, and South China.

References:

- Wan, R., Wu, G., 2009. Temporal and spatial distributions of the spring persistent rains over Southeastern China. *Journal of Meteorological Research* 23, 598e608.
- Zhang, H., Cheng, H., Cai, Y., Spötl, C., Sinha, A., Kathayat, G., Li, H., 2020. Effect of precipitation seasonality on annual oxygen isotopic composition in the area of spring persistent rain in southeastern China and its paleoclimatic implication. *Climate of the Past* 16, 211-225.
- Zhang, H., Zhang, X., Cai, Y., Sinha, A., Spötl, C., Baker, J., Kathayat, G., Liu, Z., Tian, Y., Lu, J., 2021. A data-model comparison pinpoints Holocene spatiotemporal pattern of East Asian summer monsoon. *Quaternary Science Reviews* 261, 106911.
- Zhang, H., Griffiths, M.L., Chiang, J.C., Kong, W., Wu, S., Atwood, A., Huang, J., Cheng, H., Ning, Y., Xie, S., 2018. East Asian hydroclimate modulated by the position of the westerlies during Termination I. *Science* 362, 580-583.

Reviewer #1 (Remarks to the Author):

Q1: He et al. presents a new set of transient fully coupled climate model simulations through the last deglaciation. The authors use results from these simulations to interrogate how changes in climate forcings influenced the temporal and spatial variability of water isotopes and rainfall amounts across China. What has emerged in the recent literature is that while the $\delta^{18}\text{O}$ of (in particular) speleothem records are more or less homogenous across China through the deglaciation, hydroclimate proxies suggest an antiphased response in rainfall amounts between northern and southern China in response to changes in the Atlantic Meridional Overturning Circulation (AMOC). Yet, what hasn't been so clear, until now, is how the autumn/winter monsoon varied in the past, and to what degree changes during these seasons can contribute to the annual hydroclimate variability witnessed in the proxy records. The authors find coeval changes in both summer and autumn/winter monsoon rainfall changes through the deglaciation in southern China, that is antiphased with the response in northern China. Interestingly, their model results show that this increase in autumn rainfall during Heinrich Stadial 1 and Younger Dryas was, similar to the summer monsoon, likely tied to the AMOC changes and its impact on the meridional SST gradient, which acted to "maintain the Western North Pacific Anticyclone (WPNAC) in autumn". Some caveats of this work are a lack of proxy-model comparisons, but there are a lack of high resolution winter monsoon records so this is not the fault of the authors. However, I do think the authors could have explained in some more detail how their results help to reconcile ongoing debates about what the $\delta^{18}\text{O}$ speleothem records are exactly telling us. And, are the records in northern China, for example, telling us something different from those in southern/central China? We know the $\delta^{18}\text{O}$ in the speleothems across China (and beyond) look very similar through the deglaciation. Is this due to a common mechanism related to what is happening 'upstream' of the summer monsoon moisture source? Or, is it also related to what is happening in autumn/winter, as this study shows in terms of rainfall amount. The paper is overall well written, and the interpretation of the results appear robust. Hence, I recommend this paper be published once the authors have addressed some minor comments/questions outlined below:

A1: We appreciate the overall positive comments from this reviewer. Here, the major question is the mechanism of the $\delta^{18}\text{O}_c$ and its relation to our focus here on the autumn rainfall. We reply to this comment in two parts.

First, we have now further clarified in several places in the text that, in spite of a major contribution of the autumn monsoon rainfall to the deglacial change of annual rainfall, the autumn rainfall has little contribution to the $\delta^{18}\text{O}_p$ evolution. In the revised version, we have now moved the discussion of the influence of autumn rainfall on $\delta^{18}\text{O}_p$ from the supplementary information up to the main text (see section *Influence of East Asian autumn monsoon on the speleothem $\delta^{18}\text{O}_c$*) and also to the Methods. The mechanism is explained below (as well as in the

main text). We use LGM-HS1 as the example to illustrate the underlying mechanism. The change of $\delta^{18}O_p$ is related to a change in precipitation seasonality and a change in isotopic composition. (i) A change of the rainfall seasonality is related to the change of the weight (red) in $\delta^{18}O_p = \sum_{i=1}^{12} \delta^{18}O_i \left(\frac{P_i}{P}\right)$, where i is month. In our present study, the increase in autumn precipitation weight leads to a depletion effect in $\delta^{18}O_p$ (Fig. 5e), as expected. However, this increase in autumn precipitation weight is accompanied by a weight decrease of other seasons (as the sum of the weight has to remain to be 1.), which produce an enrichment response that nearly perfectly cancels the depletion from autumn season (Fig. 5f). So the net response for the change of rainfall amount is close to zero (Fig. 5d). (ii) The change of precipitation may also change the isotopic composition (blue) in $\delta^{18}O = \sum_{i=1}^{12} \delta^{18}O_i \left(\frac{P_i}{P}\right)$. However, from LGM to HS1, this isotopic change in autumn season is also very small (Fig. 5b). As such, the total $\delta^{18}O_p$ response in autumn season is neglectable compared to that in summer monsoon as shown in He et al. 2021. A more detailed discussion on the leading role of summer monsoon contribution to the deglacial evolution of $\delta^{18}O_p$ could be found in He et al. 2021.

Second, the interpretation and implication of speleothem $\delta^{18}O_c$ has been addressed in more detail in our prior study (He et al., 2021), which reveals that the $\delta^{18}O_c$ evolution in pan-Asia monsoon regions is mostly driven by the precipitation $\delta^{18}O_p$ in summer monsoon (Fig. S4 in He et al. 2021), through the so-called “upstream effect” as well as local recycling (Fig. S6 in He et al. 2021). Accompanied with the uniform $\delta^{18}O_p$ response is a hydroclimate footprint with spatially varied precipitation response (Fig. 5A,B in He et al. 2021). This coupled $\delta^{18}O_p$ and hydroclimate are fully coherent with the change of high-level westerly jet, low-level East Asian monsoon flow, precipitation over the Indian Ocean, and their time evolution are characterized by a linear increasing trend with two negative excursions in HS1 and YD during the last deglaciation. Therefore, the uniform and widespread $\delta^{18}O_p$ evolution signal in Asian monsoon regions implies a coherent deglacial variability of the Asian summer monsoon system, even though hydroclimate varies from places to places.

Specific comments:

Q2: Line 86-87: The iTRACE model shows increased winter monsoon winds during HS1-YD coeval with the summer monsoon winds. This is interesting and indeed an important finding given that the highly-cited Yancheva et al. (2007) interpreted the opposite in their sediment record. It might benefit the reader to be made aware of this apparent contradiction between the new model results and Ti lake records. I am aware of the controversy as I am involved in this type of paleo work, though I am not sure whether the lay reader will be aware of this. This is important because these new model results shed new light on the synchronicity of the two major monsoon systems, and in my mind, settle once and for all the interpretation of this record.

A2: Thanks for pointing out this. We have added one section *Twisted relationship between EASM and EAWM* in the manuscript. Basically, the intensified EAWM wind is positively correlated with the EASM (opposite to the sense of Yancheva) if the EASM is defined in local rainfall, but is negatively correlated EASM if the EASM over South China is defined in the southerly monsoon wind over East China. Therefore, the phase relation between EASM and EAWM depends on the definition of the EASM intensity. Please refer to the section for details.

Q3: How do these results fit in with the recent model of Chiang et al. (PNAS, 2020)? Recently, Chiang et al. (2020; PNAS) demonstrated, using an isotope-equipped climate model constrained by large-scale atmospheric reanalysis fields, that year's corresponding with more enriched $\delta^{18}O$ of precipitation have a reduced summer seasonality between the months of June and October. Specifically, these authors showed that years with more enriched (depleted) $\delta^{18}O$ of precipitation have a summer-winter monsoon transition that is less (more) pronounced. The results presented in the iTRACE model are consistent with Chiang et al's results in that both show no "amount effect" in central China, but to what extent do the results here align or not align with those of Chiang et al. (2020)? Obviously, this is not a complete apples-to-apples comparison given that the He et al. manuscript is focused on the deglaciation and Chiang et al. paper is the modern climate, but I am just curious if the authors have considered this mechanism because I did not see much reference to the westerlies crossing the Tibetan Plateau, and particularly how the timing of this crossing could affect precipitation isotopes in central China during the summer-winter monsoon transition. There were hints of the jet transition hypothesis being mentioned (for example, Lines 107-110), but it just wasn't clear to me how the Chiang model fits in with these new data. Maybe I am missing something.

A3: This is an interesting question! (Chiang et al., 2020) is a modern-day study following (Chiang et al., 2015; Zhang et al., 2018). We indeed observe a south placed westerly jet on orbital and millennial timescales in the summer monsoon, along with the shift of rainbelt (supplementary Fig.1), as suggested in (Chiang et al., 2015; Chiang et al., 2020), and these shifts are associated with a summer enrichment in $\delta^{18}O_p$ in East Asia (He et al., 2021). However, this enrichment in $\delta^{18}O_p$ seems to be not mainly constrained by the position of westerly jet/rainbelt relative to Tibetan Plateau. Rather, our analysis tends to identify this enrichment related more to the upstream effect and local water vapor recycling (He et al., 2021) (also see reply A1). The reason for the different responses of $\delta^{18}O$ of deglacial (in our model) and interannual variability in "reanalysis" (Chiang et al., 2020) is worthy further exploring in the future. Our speculation at this point is that they are due to different mechanisms.

We also note that, beside the coherence with the westerly jet, the widespread $\delta^{18}O_p$ response is also highly coherent with the Asian summer monsoon systems during the last deglaciation, in particular with low level monsoon circulation in East Asia, the Pacific subtropical high, East

Asia hydroclimate, and Indian ocean rainfall (Fig. 5 in He et al. 2021). To us, this suggests that the entire Asian summer monsoon system is likely an intrinsically coupled system, from the upper level westerly jet to the low level monsoon wind and moisture transport. (Chiang et al., 2015; Zhang et al., 2018) argued that a shift of westerly jet associated with equator-to-pole upper troposphere temperature gradient causes the dipole hydroclimate response in cold stadials. (Liu et al., 2014) suggested that the low level monsoon associated with land-sea thermal contrast and subtropical high causes the dipole hydroclimate response; (He et al., 2021) further supplemented their interpretations and suggested that the South Asian monsoon and East Asian monsoon are connected by silk-road teleconnection that could shift with the westerly jet, alter the low-level monsoon flow in East Asia and yield the dipole hydroclimate response. In short, we feel, the westerly jet and the low level monsoon wind and subtropical high are different perspectives of the Asian monsoon system and their coevolution suggests that these components in the monsoon system are of high coherence. All these do lead to the ultimate question: what is the ultimate control of these coupled circulations in the Asian monsoon system?

These coherent evolutions in circulation and in turn the hydroclimate probably evolve in response to the same forcing (external to the atmosphere, e.g., solar insolation, meltwater flux), or they may be caused by the internal teleconnections forced by external forcings. For example, a change in solar insolation or meltwater flux could change meridional temperature gradient that could both affect the location of westerly jet and Pacific subtropical High linked to low-level East Asian monsoon flow (e.g., Zhang et al. 2018; Liu et al. 2014); a change in land-sea thermal contrast due to solar insolation could change the South Asian monsoon, which could in turn influence the East Asian monsoon rainfall via the silk-road teleconnection (e.g. He et al. 2021). Meanwhile, the associated upper atmospheric response also changes the westerly jet, which could also affect the silk-road teleconnection by affecting the waveguide, also contributing to the change of East Asian monsoon. Yet, it remains challenge to quantify the individual role of these circulations in driving the Asian monsoon variability and to disentangle their causal relationships. This seems to us very difficult in these type of standard model experiments, because all these aspects are tightly coupled with the monsoon system. Some highly idealized sensitivity experiments in an AGCM are needed to understand these questions in the future. In sum, we believe the community is interpreting the $\delta^{18}O_p$ and hydroclimate from different perspectives, each perspective has its pros and cons. A comprehensive view emerges when all these perspectives are considered.

Finally, turning to the autumn monsoon in present study, we suggest the WNPAC plays a critical role associated with the SST gradient on millennial timescales, which appears not directly related to the westerly jet migration, so it may be different from variabilities on interannual timescale.

By the way, it seems to us there is an inconsistency between (Chiang et al., 2020) and (Chiang et al., 2015; Zhang et al., 2018) regarding the relation between the southward shift in westerlies and

rainfall in South China: the former suggests a dry South China while the latter seems suggest a wet South China. This difference, if correct, maybe attributed to model dependence, and that interannual variability maybe different from millennial variability.

Q4: What are the implications for these results in terms of our understanding of how/why the d18O records across China are mostly synchronous through the last deglaciation despite significant changes in both summer and autumn rainfall amounts? Since the paper of Pausata et al. (2011), the general consensus in the literature has been that the d18O in speleothems reflects more regional atmospheric circulation changes, such as upstream rainout and shifting moisture sources, rather than local rainfall amount. If the model results in this manuscript are accurate, showing that autumn rainfall changes may account for a large proportion of the annual variability through HS1/YD (in addition to summer), then wouldn't this seasonal contribution of autumn rainfall also contribute to the annual d18O precipitation signal recorded in the speleothems? Is it because the patterns of both the summer and autumn rainfall changes through HS1/B-A/YD are similar, at least in central China, that you have such a homogeneous signal across the region through these events? If this is the case, then can we assume that d18O source changes (as proposed by Hu et al. 2020, albeit over orbital and not millennial timescales) may not be the main driver of the d18O changes on orbital and millennial timescales? However, as stated in the paper, there are large seasonal differences in the response of hydroclimate to changes in the AMOC between central and northern China. Thus, can these spatial and temporal changes in rainfall through the deglaciation observed in the model help to explain the speleothem d18O changes through time?

A4: Here we answer these questions point by point below.

- a. What are the implications for these results in terms of our understanding of how/why the d18O records across China are mostly synchronous through the last deglaciation despite significant changes in both summer and autumn rainfall amounts? Since the paper of Pausata et al. (2011), the general consensus in the literature has been that the d18O in speleothems reflects more regional atmospheric circulation changes, such as upstream rainout and shifting moisture sources, rather than local rainfall amount. If the model results in this manuscript are accurate, showing that autumn rainfall changes may account for a large proportion of the annual variability through HS1/YD (in addition to summer), then wouldn't this seasonal contribution of autumn rainfall also contribute to the annual d18O precipitation signal recorded in the speleothems? Is it because the patterns of both the summer and autumn rainfall changes through HS1/B-A/YD are similar, at least in central China, that you have such a homogeneous signal across the region through these events?

This question shares some similarity with Q1. The synchronicity of $\delta^{18}O_p$ across East China is due to the “upstream effect” and local water vapor recycling in the summer monsoon (see discussion in A1 and in He et al. 2021). The autumn rainfall has no major contribution to the speleothem $\delta^{18}O_p$ even though it has a major contribution on the deglacial evolution of annual rainfall. For more details, please see the discussion in A1 and section *Influence of East Asian autumn monsoon on the speleothem $\delta^{18}O_c$* in the revised manuscript. Nevertheless, this autumn monsoon change intensifies the dipole hydroclimate response in East China as suggested by observations. The $\delta^{18}O_p$ response are not affected by the rainfall pattern either as we discussed in A1 due to the cancellation in seasons.

- b. If this is the case, then can we assume that d18O source changes (as proposed by Hu et al. 2020, albeit over orbital and not millennial timescales) may not be the main driver of the d18O changes on orbital and millennial timescales? However, as stated in the paper, there are large seasonal differences in the response of hydroclimate to changes in the AMOC between central and northern China. Thus, can these spatial and temporal changes in rainfall through the deglaciation observed in the model help to explain the speleothem d18O changes through time?

This is an interesting hypothesis. As stated in A1, the deglacial change in $\delta^{18}O_p$ is contributed mainly by the summer monsoon via the upstream effect, with the autumn rainfall playing a minor role, in both the precipitation seasonality effect and isotopic composition effect. Regardless the precipitation seasonality (and pattern), the shift of water source, in principle, could lead to a uniform isotopic composition anomaly in East Asia on both orbital and millennial timescales. Modeling studies that identified the importance of the shift of $\delta^{18}O_p$ source seems to be mostly for interannual timescales (e.g., (Hu et al., 2019; Chiang et al., 2020; Kathayat et al., 2021)). However, this isotopic composition effect could be altered by a number of other factors, such as a redistribution of source water vapor to East Asia associated with a reorganization of monsoon circulation, en route depletion due to a change in Rayleigh distillation and rainout, and local condensation because of a change in the intensity of convection. Any change in these factors could lead to a widespread $\delta^{18}O_p$ response in East Asia. Our simulation supports the en route depletion effect (upstream effect) for the last deglaciation.

Q5: How do the authors reconcile the disparity in rainfall response between the YD and HS1 in northern China? The proxy (particularly lake) records by and large indicate that HS1 was drier in northern China, similar to the YD (see Fig. 3 and references therein in Zhang et al., 2018, Science). In Fig. 5B of He et al., 2021, the first MCA mode for summer rainfall clearly shows a

dipole pattern between central and northern China. Given very similar forcings between HS1 and the YD, any explanation for why the model does not reproduce the proxy results in northern China during HS1?

A5: Thanks for pointing out this question. In spite of a similar meltwater forcing, there is an important difference between LGM-HS1 and BA-YD: the former is affected additionally by significant GHG and insolation forcing (because of its longer duration), while the latter is not. We use LGM-HS1 as example in our present study and the related issues have been addressed in the supplementary discussion (*Summer monsoon rainfalls in cold stadials of HS1 and YD*). These points are stressed in our revision main context (line: 128-136). Here we further clarify it. First, proxies suggest a dry YD relative to BA, while we have not seen robust observational evidences supporting a dry HS1 relative to LGM. For example, the N. China records in Fig. 1 in (Zhang et al., 2018) does not extend to HS1. A visual examination of the rainfall proxy of Be10 (Beck et al., 2018) in N. China does not show a marked difference between HS1 and LGM (although the resolution is coarse). The lake status records do not appear to have too low temporal resolution to show the difference (Liu et al., 2014) (although we admit that we have been unable to perform an exhaustive search of literature.) Second, Fig 3 (and the experiment) in Zhang et al. 2018 would be a better analogy of BA-YD than LGM-HS1, because (i) their hosing consider neither the insolation difference nor the GHG difference. Since both forcing changes are substantial between LGM to HS1, but not between BA and YD because of the short duration (~1000 years), their simulation resembles more the BA to YD response. (ii) their simulation used PI as the hosing background, which is 5-7°C warmer than the LGM, which also makes it more relevant to BA rather than LGM.

Here in our transient simulation, North China became wet purely because of an increase in solar isolation from LGM to HS1 that overwhelms the cooling from slowdown of the AMOC (Supplementary Fig. 2d, red curve). In contrast, from BA to YD, the solar isolation remains largely unchanged, such that the hydroclimate response is driven by the slowdown of the AMOC, as suggested by Zhang et al. 2018. Indeed, even in iTRACE, N. China rainfall still is reduced relative to its rising trend between LGM to HS1 and this reduction is identified in our sensitivity experiment as caused mainly by the insolation (Supplementary Fig.2d). As also shown in our experiments, rainfall responds strongly to insolation in N. China, and weakly to meltwater, while the opposite occurs in S. China (Supplementary Fig.2).

In Fig 5B of He et al. 2021, the leading mode indeed reflects the dipole response in HS1 and YD. This occurs because the leading mode of the pan-Asian domain MCA analysis appears to be unable to identify some detailed regional features. These regional features would be represented in higher modes or in the leading mode in more regional analysis. These have been discussed in detail in section of “*Maximum Covariance Analysis (MCA) Mode*” in the supplementary information

https://advances.sciencemag.org/content/advances/suppl/2021/01/14/7.4.eabe2611.DC1/abe2611_SM.pdf).

Minor comments:

Line 30: Should be “reliable moisture proxies”

Fixed.

Figure 1a: “Frocings” should read “Forcings”

Fixed.

Reviewer #2 (Remarks to the Author):

The deglacial hydroclimate in eastern China, especially in southern China, remains in debates because of a lack of reliable paleorainfall proxy and inconsistent model simulations. Based on a deglacial simulation of an isotope-enabled Transient Climate Experiment (iTRACE) performed in the iCESM 1.3, the authors proposed that intensified hydroclimate in South China is also heavily contributed by the autumn rainfall, which are response to the slowdown of the North Atlantic thermohaline circulation (AMOC). It was suggested that the excessive rainfall in autumn results from the convergence between anomalous northerly wind due to amplified land-sea thermal contrast and anomalous southerly wind associated with an anticyclone at Western North Pacific. As far as I'm concerned, the dynamic interpretation is reasonable based on moist static energy (MSE) budget analysis. The manuscript was well written and I suggest that it can be accepted if my concerns could be addressed/considered.

We appreciate the overall positive feedback from reviewer 2.

Here are my questions for the manuscript:

Q6: In Supplementary figure 1e, we can also find a considerable amount of spring rainfall in March-May. If the authors want to use the precipitation seasonality to explain wet climate but high $\delta^{18}\text{O}$ values in southern China during the off-state of the AMOC, why the spring rainfall was not considered/discussed in this study? Based on instrumental data (1951-2010 CE), some studies suggested that the spring (MAM) rainfall amount is equivalent to the summer monsoon (JJA) rainfall amount in southeastern China, especially in the region of the “spring persistent rain” (Wan and Wu, 2009), which may contribute ~50% to amount-weighted annual $\delta^{18}\text{O}$ values (Zhang et al., 2020). A new Holocene speleothem $\delta^{18}\text{O}$ record from southeastern China also suggested that the spring rainfall may significantly influenced variations in hydrocliamte and $\delta^{18}\text{O}$ in southeastern China (Zhang et al., 2021). It will be perfect if the authors could quantify or exclude the effect of spring rainfall on the variations in both hydrocliamte and $\delta^{18}\text{O}$ in southeastern China.

A6: We have added the discussion of MAM rainfall evolution as well as the role of precipitation amount on the speleothem $\delta^{18}O_p$ in the revised manuscript (line 109-116 and section *Influence of East Asian autumn monsoon on the speleothem $\delta^{18}O_c$*). Overall, the magnitude of MAM rainfall response is significantly smaller than those in JJA and SON, and it is highly linked to the southerly monsoon flow driven by the orbital forcing during the last deglaciation.

Spring rainfall does not have substantial impact on $\delta^{18}O_p$ evolution either. The change of $\delta^{18}O_p$ is composed of a change in precipitation seasonality and isotopic composition. The precipitation seasonality of spring and autumn are not important in shaping the speleothem $\delta^{18}O_p$ when considering the net annual effect, as the change of precipitation weight is strictly equal to 0 $\sum_{i=1}^{12} \Delta \left(\frac{P_i}{P} \right) = 0$. Any positive (negative) change in a particular season would be compensated by a negative (positive) change in other seasons, such that the net effect is close to zero in HS1 (Fig. 5d) and the whole last deglaciation (Fig. S4 in He et al. 2021). The isotopic composition effect in MAM shows some responses, which is mostly contributed by May associated with the broadly-defined East Asian summer monsoon (He et al., 2021), and this response is much weaker than the non-MAM season (Fig.R1)

Fig. R1. $\delta^{18}O_p$ response between HS1 and LGM (HS1 - LGM). a, annual isotopic composition effect. b-c, as in a but for portions of MAM and non-MAM. d, annual precipitation seasonality effect, e-f, as in d, but for portions of MAM and non-MAM. The red contour in b is isotopic composition effect in May.

Q7: From the full manuscript, the main experiments and evidences were obtained on the basis of differences between H1 and LGM. I am curious about the difference between YD and BA, which may better show the hydroclimate changes during the last deglaciation. Please point out it if the authors think it's not necessary to show/analysis the differences between the YD and BA or the comparison between H1 and LGM is enough.

A7: Good question. This question is partially answered in the last comment from reviewer 1. We specifically employ LGM-HS1 as example in our present study because we would like to supplement the discussion about the wetness in North China in HS1, which puzzles readers (also reviewer 1) in He et al. 2021.

The BA-YD is indeed better than the LGM-HS1 for the illustration of the impact of meltwater, because the transition duration is short (~1000 years) and therefore the effects of greenhouse gas and orbital forcing are minimum in this transition. However, based on the leading mode of the MCA, the precipitation in South China in BA-YD is quite similar to the LGM-HS1, except for the magnitude, as reflected by the PCs and precipitation intensity in Fig. 2b. Here we also show the precipitation and circulation response in Fig. R2. In this revised version, we also mentioned that no essential difference exists between BA-YD and LGM-HS1 in line 159. Also see reply A5 to reviewer 1 for additional discussions.

Fig. R2. Climatology difference between YD and BA (YD-BA) in iTRACE. Shading: precipitation; Vector: low-level atmospheric circulation. It is clear that the WNPAC and rainfall anomaly in South China are quite similar to that in HS1, but with a weaker magnitude as suggested by the PCs in Fig.2b.

Q8: A tripole pattern of summer monsoon rainfall (dry-wet-dry) over eastern China can be recognized in the observations and model simulations (e.g., Zhang et al., 2018), and the Haozhu cave is located in the middle-lower reaches of the Yangtze River Valley (~35°N). It was suggested that a wetter central eastern China during North Atlantic cooling episodes (the weak or off-state of the AMOC) based on the multiproxy speleothem record of Haozhu cave and model simulation (e.g., Zhang et al., 2018). In this manuscript, the eastern China was divided into North (37-50°N, 108-120°E) and South (20-35°N, 108-120°E) China, in other words, a dipole pattern of rainfall in eastern China. In addition, from the supplementary figure 1e, we do see the autumn rainfall is equivalent to the summer rainfall at the latitude of Haozhu cave (~35°N), however, there is high spring rainfall but low autumn rainfall in the region of 20-28°N. How to reconcile

this differences? The authors may reconsider the division of eastern China as North and South China or North, Central-east, and South China.

A8: We appreciate the suggestion. The sandwich-pattern hydroclimate response in East Asian summer monsoon appears dominant on interannual timescales (Chiang et al., 2017), especially as the direct response to El Nino (Wen et al., 2019). On millennial timescales, the response pattern appears to be dominated by a dipole pattern, as suggested by observations and modeling studies (Zhang et al., 2018; Chiang et al., 2015). This dipole pattern is also observed in (He et al., 2021), in which the leading MCA mode accounts for more than 90% of the total variance. Indeed, even for the present day observation, the dipole pattern seems to become dominant for interdecadal variability (Ding et al., 2008) and for the ~50-year trend (Liu et al., 2014). The different response pattern at different time scales are possible because they are forced by different physical mechanisms. In our study, we further suggest that this dipole pattern is intensified by the autumn monsoon rainfall variability in South China. Here, we also divide the East China into three regions (North China, Central China, and South China) and plot the annual precipitation evolution to confirm the dipole response during the last deglaciation (Fig. R3). Clearly, in millennial events like YD, when there is little orbital effect involved, it shows a dipole hydroclimate response in East China. While in HS1, the both North China and South China becomes wet due to the orbital and GHG forcing as we discussed above. The wetness in North China, however, is due to the trend associated with orbital/GHG that cancels the drying due to melting water, as can be seen in its time evolution that shows a deficient rainfall relative to the trend.

Regarding the MAM rainfall in Supplementary Fig. 1e, the intensification of precipitation in HS1 in 20-28°N(South China) is mostly achieved in May (Fig. R4), which is closer to a pre-meyu rainfall in the summer monsoon. The strictly defined spring persistence rainfall (March-April) is close to 0, smaller than the autumn rainfall (Fig. R4).

Fig. R3. Annual mean precipitation evolution in East Asian (108-120°E). The East Asian has been divided into 3 regions, namely South China (20-30°N), Central China (30-37°N), and North China (37-50°N). A 9-decadal running mean has been applied on each curve for presentation.

Fig. R4 Spring and Autumn precipitation anomaly (HS1-LGM) in South China (20-28°N; 108-120°E). The intensified precipitation in MAM is mainly due to the rainfall in May that is closer to the pre-meiyu in summer monsoon.

References

- Beck, J.W., W. Zhou, C. Li, Z. Wu, L. White, F. Xian, X. Kong, and Z. An, 2018: A 550,000-year record of East Asian monsoon rainfall from ^{10}Be in loess. *Science*, **360**, 877–881.
- Chiang, J.C.H., L.M. Swenson, and W. Kong, 2017: Role of seasonal transitions and the westerlies in the interannual variability of the East Asian summer monsoon precipitation. *Geophysical Research Letters*, **44**, 3788–3795.
- Chiang, J.C.H., I.Y. Fung, C.H. Wu, Y. Cai, J.P. Edman, Y. Liu, J.A. Day, T. Bhattacharya, Y. Mondal, and C.A. Labrousse, 2015: Role of seasonal transitions and westerly jets in East Asian paleoclimate. *Quaternary Science Reviews*, **108**, 111–129.
- Chiang, J.C.H., M.J. Herman, K. Yoshimura, and I.Y. Fung, 2020: Enriched East Asian oxygen isotope of precipitation indicates reduced summer seasonality in regional climate and westerlies. *Proceedings of the National Academy of Sciences*, **117**, 14745–14750.
- Ding, Y., Z. Wang, and Y. Sun, 2008: Inter-decadal variation of the summer precipitation in East China and its association with decreasing Asian summer monsoon. Part I: Observed evidences. *International Journal of Climatology: A Journal of the Royal Meteorological Society*, **28**, 1139–1161.
- He, C., Z. Liu, B.L. Otto-Bliesner, E.C. Brady, C. Zhu, R. Tomas, P.U. Clark, J. Zhu, A. Jahn, and S. Gu, 2021: Hydroclimate footprint of pan-Asian monsoon water isotope during the last deglaciation. *Science Advances*, **7**, eabe2611.
- Hu, J., J. Emile-Geay, C. Tabor, J. Nusbaumer, and J. Partin, 2019: Deciphering oxygen isotope records from Chinese speleothems with an isotope-enabled climate model. *Paleoceanography and Paleoclimatology*, **34**, 2098–2112.
- Kathayat, G., A. Sinha, M. Tanoue, K. Yoshimura, H. Li, H. Zhang, and H. Cheng, 2021: Interannual oxygen isotope variability in Indian summer monsoon precipitation reflects changes in moisture sources. *Communications Earth & Environment*, **2**, 1–10.
- Liu, Z., X. Wen, E.C. Brady, B. Otto-Bliesner, G. Yu, H. Lu, H. Cheng, Y. Wang, W. Zheng, Y. Ding, R.L. Edwards, J. Cheng, W. Liu, and H. Yang, 2014: Chinese cave records and the east asia summer monsoon. *Quaternary Science Reviews*, **83**, 115–128.
- Wen, N., Z. Liu, and L. Li, 2019: Direct ENSO impact on East Asian summer precipitation in the developing summer. *Climate dynamics*, **52**, 6799–6815.
- Zhang, H., M.L. Griffiths, J.C.H. Chiang, W. Kong, S. Wu, A. Atwood, J. Huang, H. Cheng, Y. Ning, and S. Xie, 2018: East Asian hydroclimate modulated by the position of the westerlies during Termination I. *Science*, **362**, 580–583.

REVIEWERS' COMMENTS

Reviewer #1 (Remarks to the Author):

I commend the authors for their thorough responses and detailed amendments to their manuscript. I am satisfied with the authors' revisions and therefore can now recommend the paper be accepted for publication in Nature Communications. I only have some minor editorial suggestions as indicated in the attached annotated manuscript.

Reviewer #2 (Remarks to the Author):

All of my concerns have been addressed/considered in the new version. I think this is an important modelling study that provides details how do the spring, summer and autumn rainfall change and their contributions to annual precipitation $\delta^{18}O$ and thus speleothem $\delta^{18}O$ proxy during the last deglaciation.

Reviewer #1 (Remarks to the Author):

I commend the authors for their thorough responses and detailed amendments to their manuscript. I am satisfied with the authors' revisions and therefore can now recommend the paper be accepted for publication in Nature Communications. I only have some minor editorial suggestions as indicated in the attached annotated manuscript.

We appreciate this reviewer's comments, which improved our manuscript substantially. Typos have been fixed.

Reviewer #2 (Remarks to the Author):

All of my concerns have been addressed/considered in the new version. I think this is an important modelling study that provides details how do the spring, summer and autumn rainfall change and their contributions to annual precipitation $\delta^{18}O$ and thus speleothem $\delta^{18}O$ proxy during the last deglaciation.

We appreciate this reviewer's comments, which improved our manuscript substantially.